# FROM BIAS TO BENEFIT: PLACE GOOD DOCUMENTS IN GOOD POSITIONS

## ABSTRACT

Large language models (LLMs) exhibit a U-shaped positional bias in processing input information, characterized by heightened attention to tokens at the beginning and end of the prompt while ignoring information in the middle, also known as the Lost-in-the-Middle phenomenon. In this paper, we investigate the internal mechanisms underlying this phenomenon by analyzing how positional bias influences attention weights across both horizontal (input-level) and vertical (layer-level) dimensions of the model. Based on these findings, we propose U-shaped Placement, a strategy that leverages inherent positional bias of the model by assigning documents to positions that align with its attention pattern. By combining this placement strategy with the importance estimations of documents, effectively placing good documents in good positions, we enhance the model's ability to utilize documents within two iterations. Experimental results demonstrate that our method consistently outperforms existing baselines across multiple models and datasets, indicating that leveraging positional bias can bring improved document utilization capability. Our codes are submitted with the paper and will be publicly available.

## 1 INTRODUCTION

As large language models(LLM) continue to evolve, they have achieved superior performance in many tasks, especially in Question Answering (QA) tasks (Touvron et al., 2023; Achiam et al., 2023; DeepSeek-AI, 2025). Furthermore, Retrieval Augmented Generation(RAG) has become a widely recognized paradigm by supplementing the model with external knowledge in the form of context, which helps to improve the factual accuracy and reliability of the answers (Gao et al., 2023; Asai et al., 2023). However, the quality of input documents is variable (Shi et al., 2023; Yoran et al., 2024; Wu et al., 2024) due to the inadequate performance of the retriever (Yan et al., 2024) or the alignment gap between the retriever and the generator (Ke et al., 2024; Li & Ramakrishnan, 2025).

How to improve a model's ability to utilize documents with inputs of varying quality is a challenging and realistic research topic, and this is also part of the model robustness problem (Shi et al., 2023; Yoran et al., 2024; Zhou et al., 2025). Previous works improve the robustness of the model by incorporating irrelevant and interfering documents into the supervised fine-tuning process (Pan et al., 2024; Yoran et al., 2024; Tu et al., 2025), which is customized and requires additional training resources. Instead of direct training, we focus on the model's properties of prompt utilization, especially the ability to leverage documents in different positions.

The retrieval results convey the relative importance of documents through their ranking and order in the prompt (Gao et al., 2023). But language models exhibit a U-shaped positional bias in processing input information, assigning greater weight to content at the beginning and end of the prompt while often ignoring content in the middle. This phenomenon was initially identified in Liu et al. (2024) and later corroborated through performance evaluations in RAG tasks by Cuconasu et al. (2024) and Wu et al. (2024). However, research on the underlying mechanisms of this U-shaped curve, as revealed through the model's internal states, remains limited.

In this paper, we first analyze the positional bias towards documents by examining the internal mechanisms of LLMs. We assess the influence of document position on attention weights from both horizontal (input-level) and vertical (layer-level) perspectives, using systematically constructed inputs and probing different layers of the model. The value of attention weight not only captures document relevance but also encapsulates the influence of positional bias (Peysakhovich & Lerer,

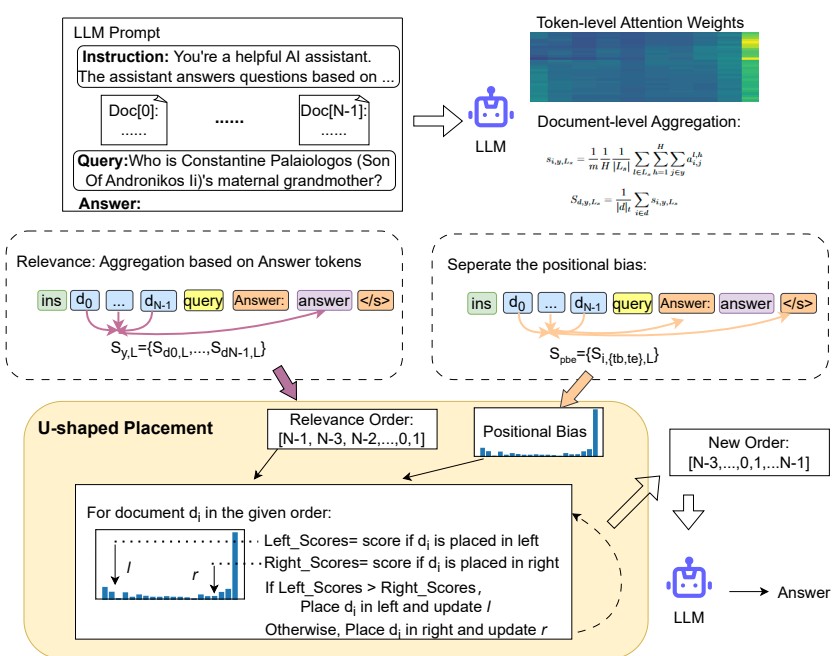

Figure 1: The framework of our proposed pipeline.

2023; Chen et al., 2024; Liu et al., 2025). Our analysis isolates the positional effect and confirms that it consistently follows the U-shaped curve.

Building upon these insights, we propose U-shaped Placement, a strategy that reorganizes documents to align with the model's inherent positional bias. This approach is integrated into a two-round iterative generation process that refines input prompts based on the model's internal states, as depicted in Figure 1. Specifically, during the first round, we compute document importance scores using attention weights and simultaneously estimate the model's positional bias. These two signals are then combined to rearrange documents for the second iteration, ensuring that content deemed most relevant is placed in positions that receive higher attention. We conduct comprehensive experiments on several multi-document QA datasets utilizing various commonly used LLMs, demonstrating that our method consistently outperforms baselines and yields higher response quality, indicating that leveraging positional bias can bring improved document utilization capability. Our method requires no additional training and can be readily applied to different models and datasets.

Our contributions are as follows: (1) We study the influence of document position on attention weights from both horizontal (input-level) and vertical (layer-level) perspectives, revealing the internal mechanisms underlying the Lost-in-the-Middle phenomenon. (2) We propose a novel strategy called U-shaped Placement to take advantage of inherent positional bias in the generation process, which is the first to our knowledge.(3) Comprehensive experiments show that our method can improve the effectiveness of document utilization in a training-free manner.

## 2 RELATED WORKS

### 2.1 RETRIEVAL AUGMENTED GENERATION

Retrieval-augmented generation (RAG) has exhibited significant effectiveness in addressing issues such as hallucinations by introducing external knowledge into context or training objectives (Gao et al., 2023; Asai et al., 2023; Tu et al., 2025; Luo et al., 2024; 2025). However, irrelevant and distracting information can adversely affect the generated results (Shi et al., 2023; Yoran et al., 2024; Wu et al., 2024). Previous work has explored various improvement strategies, such as improving the retriever (Shi et al., 2024), designing new rerankers (Kim & Lee, 2024), investigating gaps between

the retriever and generator (Ke et al., 2024; Li & Ramakrishnan, 2025), and improving the robustness of the LLM, especially interference resistance (Xiang et al., 2024; Yoran et al., 2024). Rather than directly adding documents to the training or supervised fine-tuning process (Pan et al., 2024; Yoran et al., 2024; Tu et al., 2025), we study the internal utilization characteristic of documents at different positions and dynamically modify inputs based on positional bias to improve robustness.

## 2.2 DOCUMENT RELEVANCE

In RAG pipeline, the external retriever or reranker will give a relevance score, and the different importance would be reflected mainly by the positional order in prompt, rather than the value itself (Gao et al., 2023; Shi et al., 2024; Kim & Lee, 2024). Including the relevance score into the prompts may affect the generated results (Pan et al., 2024), but this requires a high level of the instruction-following ability. In addition to utilizing externally given relevance scores, there are also some works that let the model itself give a judgment on the relevance of documents through prompt engineering (Qin et al., 2024; Sun et al., 2023; Niu et al., 2024), adding probing structures (Baek et al., 2024; Wang et al., 2024), or internal attention weight (Peysakhovich & Lerer, 2023; Chen et al., 2024; Liu et al., 2025). We also use attention weights as the basis for model importance estimation for documents, but we compute them differently and further combine them with positional bias to optimize the inputs.

## 2.3 POSITIONAL BIAS

The LLMs are unable to treat the information in the prompt equally and have a positional bias, which is part of the model's prompt-sensitive properties (Xie et al., 2024). It tends to pay more attention to information at the beginning and the end, and to ignore those in the middle, which is characterized by a U-shape curve. This "Lost in the Middle" phenomenon was first identified in Liu et al. (2024). To date, many RAG and long text-related works (Cuconasu et al., 2024; Wu et al., 2024; Xu et al., 2024) have investigated this issue by showing the performance difference caused by positional bias. We study this phenomenon from internal attention weight of the LLM both horizontally and vertically, offering a new perspective to investigate the U-shaped positional bias, and we propose a method to take advantage of positional bias during the generation process.

# 3 INVESTIGATION ON POSITIONAL BIAS

In this section, we investigate the model's positional bias and relevance assessment toward documents placed at different prompt locations. Building upon empirical performance variations observed across positions, we further analyze these behaviors through the model's internal states, with a particular focus on attention weights.

## 3.1 NOTATIONS

We formulate the task as generating the answer based on a given question and retrieved documents, following standard RAG settings. For each sample, we use $q$ to present the question. The retrieval documents are denoted as $D = \{d_0, d_1, ..., d_{N-1}\}$, where $d_i$ is a single document, and $N$ is the total number of documents. $x = \{x_0, x_1, ..., x_{k-1}\}$ is the input of large language models, where $k$ is the number of tokens contained in the input, i.e., the token length. The input $x$ is constructed based on $q$, $D$, and a certain prompt template $T$. And the output answer is indicated as $y = \{y_0, y_1, ..., y_{m-1}\}$, where $m$ is the token length of $y$. The language model is presented as $\theta$ and generates each token in $y$ with auto-regressive style.

## 3.2 PRELIMINARY EXPERIMENTS

To demonstrate the effect of position, we first compare model performance under two standard configurations: unordered documents and documents ordered by external relevance. These settings represent common practices in both RAG evaluations and practical scenarios.

**Datasets** We apply the datasets processed by Pan et al. (2024), which include both randomized (denoted as *Unordered*) and relevance-ordered (denoted as *Ordered*) versions to minimize

processing-related randomness. Due to computational constraints, our experiments are conducted on three widely-used open-domain multi-document QA benchmarks: HotpotQA (Yang et al., 2018), Musique (Trivedi et al., 2022), and 2WikiMHQA (Ho et al., 2020). The details of datasets can be found in the original paper or Appendix B.1.

**Models and Metrics**   We test four popular open-source LLMs: Vicuna-7B (Chiang et al., 2023), Llama-3.1-8B (Dubey et al., 2024), Qwen2.5-7B and Qwen2.5-7B-Instruct (Yang et al., 2024). We follow Pan et al. (2024) and utilize Exact Match (EM) as the primary evaluation metric, which checks whether the short answers provided are exact substrings of the generation.

**Implementation**   Hyperparameters including temperature and instruction format remain consistent with their setup. Unlike their work, however, we conduct experiments under a zero-shot setting to better isolate and examine the model's intrinsic positional bias and relevance assessment mechanisms. The placement of documents in the prompt adheres to Pan et al. (2024), positioning the most relevant documents closest to the question when documents are ordered. Previous researches (Cuconasu et al., 2024; Liu et al., 2025) have also confirmed that this placement is a widely applied paradigm and strong baseline. Additional details regarding prompt templates and implementation are provided in Appendix B.2.

Table 1:   Original zero-shot model performance in HotpotQA(H), Musique(M) and 2WikiMHQA(W) datasets of CAGB benchmark (Pan et al., 2024).

| Prompt | Vicuna-7b | | | Llama-3.1-8b | | | Qwen2.5-7b | | | Qwen2.5-7b-ins | | |
|---|---|---|---|---|---|---|---|---|---|---|---|---|
| | H | M | W | H | M | W | H | M | W | H | M | W |
| Unordered | 0.392 | 0.238 | 0.452 | 0.292 | 0.192 | 0.35 | 0.376 | 0.298 | 0.39 | 0.468 | 0.458 | 0.48 |
| Ordered | 0.4 | 0.312 | 0.482 | 0.302 | 0.238 | 0.358 | 0.408 | 0.342 | 0.41 | 0.472 | 0.5 | 0.526 |

The results of the EM values are presented in Table 1. The observed inconsistency with the results reported in Pan et al. (2024) can be primarily attributed to our use of a zero-shot evaluation setting, along with potential discrepancies in huggingface versions and hardware configurations. The results indicate that, although performance varies across models and datasets, all models are affected by document position and unable to utilize information from each position equally, confirming the prevalence of positional bias. As the number of documents increases, the effect of position becomes more pronounced. This is clearly demonstrated by the Musique dataset, which contains 20 documents and exhibits substantially greater sensitivity to ordering changes.

To conduct a more fine-grained study of the varying effects at different positions, we explore the dependency of the generated answer token on the context documents using the metric developed by Qi et al. (2024), called MIRAGE. Since this metric also analyzes the generated results and is not the focus of this paper, we provide the corresponding experimental details and results in the Appendix C. These results similarly indicate the existence of positional bias, showing that the model relies more on documents placed at the beginning and end of the input.

## 3.3 ATTENTION WEIGHT

Both downstream performance and the MIRAGE metric reflect the presence of positional bias. In this section, we investigate the underlying mechanisms and internal states of the model. Attention weights capture, at the token level, the influence of context tokens on answer tokens during generation. To assess this influence at the document level, we aggregate attention weights as follows:

$$s_{i,y,L_s} = \frac{1}{m} \frac{1}{H} \frac{1}{|L_s|} \sum_{l \in L_s} \sum_{h=1}^{H} \sum_{j \in y} a_{i,j}^{l,h} \tag{1}$$

$$S_{d,y,L_s} = \frac{1}{|d|_t} \sum_{i \in d} s_{i,y,L_s} \tag{2}$$

where $a_{i,j}^{l,h}$ denotes the attention weight from the token $i$ (from the document $d$ whose token length is $|d|_t$) to the token $j$ (from the answer $y$ whose token length is $m$) by the attention head $h$ at layer

$l$, $H$ is the total number of attention heads, and $L_s$ is the set of selected layers. After obtaining the influence score of each token in the document on the answer at token level ($s_{i,y,L_s}$), we then aggregate and normalize by removing the influence of length to obtain attention weight value at document level for the answer ($S_{d,y,L_s}$). $S_{y,L_s} = \{S_{d_0,y,L_s}, ..., S_{d_{N-1},y,L_s}\}$ is the overall set of document scores.

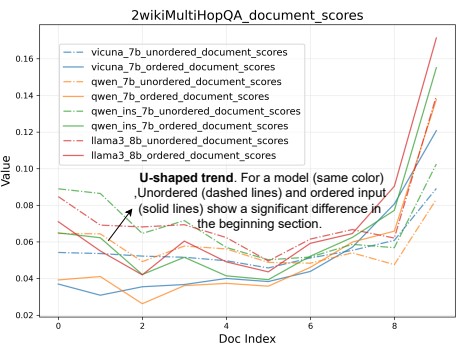

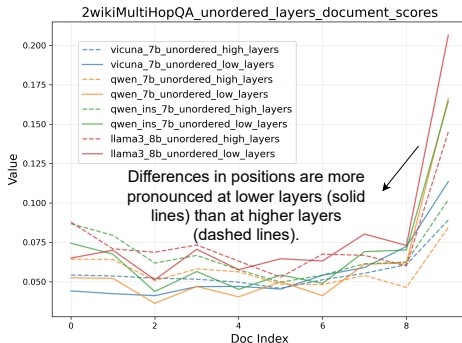

(a) Horizontal analysis: document scores $S_{y,L_{all}}$ under unordered and ordered input.

(b) Vertical analysis: document scores $S_{y,L_s}$ with different selected layers under unordered input.

Figure 2: The document scores of all models on 2wikiMultiHopQA datasets. The results of the same model are shown in the same color.

We then analyze the effect of position on document-level scores from both horizontal and vertical perspectives. Horizontally, we compare document scores across different positions under varying input conditions. We first set $L_s$ to all layers and calculate $S_{y,L_{all}}$. To ensure robustness, we randomly sample 50 instances from the 2WikiMultiHopQA dataset and average the results for clearer visualization, as shown in Figure 2a. Complete results across all models and datasets are provided in Appendix D.1. The results indicate that document scores $S_{y,L_{all}}$ exhibit a U-shaped distribution across positions under both ordered and unordered input conditions. However, under ordered input, the U-shaped curve is more skewed toward the end of the input (closer to the question), displaying a steeper profile. In contrast, the U-shape under unordered input is gentler, with less pronounced disparities between the beginning and the end portion. These findings suggest that the model's internal estimation of document importance is influenced by positional bias in a U-shaped manner, and the extent of this bias varies with the ordering of input documents.

From a vertical perspective, we further examine the effect of different layer selections under the same prompt. We partition all layers into lower and higher halves and compare the document scores derived from each group. To clearly illustrate the U-shaped positional bias with minimal interference from document relevance, we present results using unordered inputs on the 2WikiMultihopQA dataset in Figure 2b, as unordered inputs make positional bias more evident and standard than ordered inputs. Complete results are available in Appendix D.2. The results indicate that although the absolute values of document scores differ between the lower and higher layers, both exhibit a similar U-shaped trend across positions. Notably, positional distinctions are more pronounced in the lower layers. This observation aligns with the widely accepted view that lower layers are more sensitive to positional information, while higher layers focus on processing semantic content.

## 4 SEPARATE AND UTILIZE POSITIONAL BIAS

The attention weight reflects the overall influence of context tokens on answer tokens, including semantic relevance and positional influence. How to directly obtain the positional influence in the generation process is a problem worth studying.

While prior work has employed meaningless queries to study attention patterns (Chen et al., 2024), this approach necessitates an additional LLM call and focuses on query tokens rather than answer tokens. In our method, we aggregate attention weights corresponding to the token immediately preceding the answer and the terminating token (highlighted in orange in Figure 1). This choice is motivated by the observation that the token preceding the answer (e.g., "is" or ":") typically

carries little semantic information. By integrating attention scores from both the beginning and the end of the answer, we construct a composite representation of the overall positional characteristics associated with the answer tokens. This strategy aligns with the intuitive notion that combining start and end positional cues can effectively approximate the holistic positional information.

As lower layers have been shown to capture positional signals more explicitly, we perform this aggregation over a selected set of lower layers, denoted as $L_l$. We visualize the resulting scores $S_{\{t_b,t_e\},L_l}$ for the 2WikiMultihopQA dataset in Figure 3. Results for other datasets are provided in Appendix D.3. In contrast to the document-level scores shown in Figure 2a, these positional scores exhibit no significant variation between unordered and ordered inputs. This suggests that our aggregated representation effectively captures general positional characteristics of the answer, largely independent of document ordering.

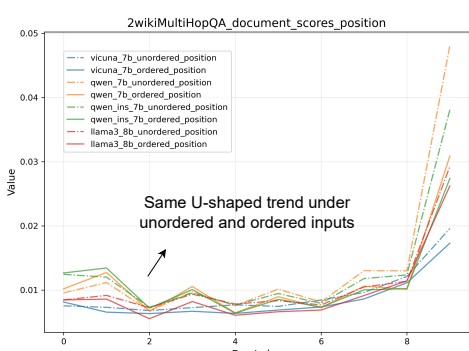

Figure 3: The positional scores $S_{\{t_b,t_e\},L_l}$ on 2wikiMultiHopQA datasets.

After separating the positional bias, we hope to use it to improve the model's ability to utilize documents. A straightforward strategy would be to rank documents directly according to the aggregated score $S_{\{t_b,t_e\},L_l}$, from most to least preferred. However, this approach encounters a practical issue related to length variability. Since $S_{\{t_b,t_e\},L_l}$ is length-normalized, the actual token capacity associated with each score may vary significantly. For example, the position with the highest score may only contain 100 token positions, and placing a document with more than 100 tokens will use the positions corresponding to other scores. To address this issue, we operate directly on token-level scores $s_{i,\{t_b,t_e\},L_l}$ rather than document-level aggregates. The underlying intuition remains consistent: to place high-quality documents in positions that receive higher attention. Concretely, we propose an allocation algorithm that considers documents in descending order of relevance and uses the U-shaped attention profile to place each document to either the beginning or the end of the available prompt space. At each step, the algorithm evaluates whether placing the document on the left (beginning) or right (end) of the remaining context yields a higher token-level score, and assigns it accordingly. This process continues until all documents are placed, resulting in a U-shaped arrangement that aligns with the model's inherent attention bias. The complete procedure, termed U-shaped Placement, is formalized in pseudocode in Algorithm 1.

---

**Algorithm 1** U-shaped Placement

---

**Input** Relevance ranking $R$, attention weight $A_\theta$, preceding token $t_b$ and terminating token $t_e$ of answer, the collection of token lengths for all documents $T_l = \{T_0, ..., T_{N-1} | T_i = |d_i|_t\}$.
 1: Ensure that the relevant ones in $R$ come first;
 2: Get $S_{pbe} = \{s_{i,\{t_b,t_e\},L_l}\}$ based on $A_\theta$; ▷ equation 1
 3: Initialization: $l = 0, r = \sum_{i=0}^{N-1} T_i, l_{idx} = 0, r_{idx} = N - 1, R_u = [0] * N$;
 4: **for** i $\in R$ **do**
 5:     $T_i = T_l[i]$;
 6:     Left_Scores = $S_{pbe}[l : l + T_i]$.sum();
 7:     Right_Scores = $S_{pbe}[r - T_i : r]$.sum();
 8:     **if** Right_Scores $\geq$ Left_Scores **then**
 9:         $R_u[r_{idx}] = i, r_{idx} = r_{idx} - 1, r = r - T_i$;
10:     **else**
11:         $R_u[l_{idx}] = i, l_{idx} = l_{idx} + 1, l = l + T_i$;
12:     **end if**
13: **end for**
**Output:** The new ranking $R_u$

---

The U-shaped Placement approach can be combined with all kinds of document ranking methods. We combine it with the previously obtained document scores that are aggregated based on the answer tokens, and modify the inputs for the next round, thus improving the overall ability of the model to

utilize documents. The complete pipeline is summarized in pseudocode in Algorithm 2, where $L_l$ and $L_h$ denote the lower and higher halves of the model layers, respectively.

---

**Algorithm 2** Place Good Documents in Good Positions

---

**Input** Prompt Template $T$, LLM $\theta$, Question $q$, Documents $D = \{d_0, ..., d_{N-1}\}$.
 1: Construct input: $x = T(q, D)$;
 2: Get the output from LLM: $y, A_\theta = \theta(x)$;
 3: Calculate $S_{y,L_h}$; $\triangleright$ equation 2
 4: Rank the documents based on $S_{y,L_h}$ as $R_a$;
 5: Get token lengths of each document from $x$ to construct $T_l$, and locate $t_b$ and $t_e$ from $x$;
 6: $R_u$= U-shaped Placement($R_a, A_\theta, t_b, t_e, T_l$);
 7: Reconstruct the input based on $R_u$: $x_u = T(q, R_u(D))$;
 8: Get the final answer: $y = \theta(x_u)$
**Output:** The output answer $y$

---

The algorithm is essentially two rounds of iterations of the LLM, using the attention weight from the first round to obtain the model's ranking of the documents and positional bias, and then placing good documents in good positions according to the U-shape, and reconstructing the inputs to generate the final answer in the second round.

## 5 EXPERIMENTS

### 5.1 BASELINES

The basic setting of the experiment is the same as preliminary experiments in section 3.2, including the datasets, models, metrics, and so on.

Our work is essentially a two-round iteration of the LLM, so we mainly consider similarly set-up baselines for fair comparison, and the following is a brief description of the baselines we consider: **(1)Vanilla**: The most basic baseline, generating answers directly based on inputs. **(2)RankGPT** (Sun et al., 2023): Two rounds of iteration, the first round uses the model to sort the documents in listwise style and the second round generates the answer. The prompt template used in the first round is shown in the Appendix F. **(3)Attention Sorting** (Peysakhovich & Lerer, 2023): Two rounds of iteration, average per-document attention is computed for the first generated token in the first round, and then documents are sorted based on the attention scores for the second round. **(4)ICR** (Chen et al., 2024):Two rounds of iteration, the first round aggregates the contextual attention weight corresponding to all query tokens and calibrates it with the meaningless query to get the document order, and the second round generates the answers based on the reordered document. **(5)SELFELICIT** (Liu et al., 2025): Two rounds of iteration, average per-sentence attention is computed for the first generated token in the first round and then important sentences are selected to be emphasized with special token in the input for the second round.

### 5.2 MAIN RESULTS

The results are shown in Table 2. The results show that: (1) Our method outperforms previous baselines on most datasets and models, under both unordered and ordered input settings. (2) Improvements are more substantial under unordered inputs than under ordered inputs. This can be partly attributed to the greater potential for enhancement in unordered settings. Notably, our approach applied to unordered inputs can surpass the performance of the vanilla ordered baseline that relies on external retrieval rankings, demonstrating its ability to infer an effective document order even without prior ranking. The gains under ordered inputs further confirm that our method enhances the model's capacity to utilize documents effectively. (3) Improvements are more pronounced on datasets with more documents, such as Musique. And our method can be applied to experiments involving any number of documents, which is proved in section 6.3. In terms of models, greater gains are observed on the Qwen series compared to Vicuna-7b, which may be related to the base capability of the model: stronger models provide more reliable internal state signals. Nevertheless, our method delivers consistent performance improvements across diverse models and datasets.

Table 2: Zero-shot model performance in HotpotQA(H), Musique(M) and 2WikiMHQA(W) datasets of CAGB benchmark (Pan et al., 2024). See section 5.1 for more details on baselines.

| Prompt | Methods | Vicuna-7b | | | Llama-3.1-8b | | | Qwen2.5-7b | | | Qwen2.5-7b-ins | | |
|---|---|---|---|---|---|---|---|---|---|---|---|---|---|
| | | H | M | W | H | M | W | H | M | W | H | M | W |
| Unordered | Vanilla | 0.392 | 0.238 | 0.452 | 0.292 | 0.192 | 0.35 | 0.376 | 0.298 | 0.39 | 0.468 | 0.458 | 0.48 |
| | RankGPT | 0.401 | 0.22 | 0.46 | 0.278 | 0.196 | 0.334 | 0.372 | 0.343 | 0.402 | 0.466 | 0.51 | 0.526 |
| | AttentionSort | 0.386 | 0.291 | 0.484 | 0.296 | 0.216 | 0.336 | 0.398 | 0.375 | 0.4 | 0.462 | 0.495 | 0.484 |
| | ICR | **0.421** | 0.305 | 0.498 | 0.298 | 0.237 | 0.368 | 0.379 | 0.385 | 0.384 | 0.469 | 0.511 | 0.522 |
| | SELFELICIT | 0.378 | 0.257 | 0.462 | **0.308** | 0.229 | 0.359 | 0.393 | 0.298 | 0.43 | 0.417 | 0.311 | 0.382 |
| | Our | 0.414 | **0.31** | **0.506** | 0.302 | **0.245** | **0.372** | **0.402** | **0.393** | **0.434** | **0.482** | **0.513** | **0.536** |
| Ordered | Vanilla | 0.4 | 0.312 | 0.482 | 0.302 | 0.238 | 0.358 | 0.408 | 0.342 | 0.41 | 0.472 | 0.5 | 0.526 |
| | RankGPT | 0.403 | 0.28 | 0.502 | 0.284 | 0.21 | 0.342 | 0.391 | 0.358 | 0.416 | 0.493 | 0.516 | 0.53 |
| | AttentionSort | 0.404 | 0.3 | 0.474 | 0.294 | 0.218 | 0.342 | 0.408 | 0.385 | 0.432 | 0.492 | 0.485 | 0.518 |
| | ICR | 0.413 | 0.307 | **0.512** | 0.301 | 0.254 | 0.353 | 0.406 | 0.378 | 0.4 | 0.487 | 0.537 | 0.504 |
| | SELFELICIT | 0.393 | 0.314 | 0.482 | **0.314** | 0.261 | 0.372 | 0.411 | 0.342 | 0.432 | 0.417 | 0.447 | 0.372 |
| | Our | **0.426** | **0.315** | 0.51 | 0.304 | **0.267** | **0.378** | 0.412 | **0.401** | **0.436** | **0.495** | **0.555** | **0.542** |

# 6 ANALYSIS

The pipeline comprises two key components: deriving document order and positional bias from the model's internal states. While the combined effect of these components has been validated in the main experiments, this section examines their individual contributions. Some additional attempts are presented in Appendix G.

## 6.1 THE INFLUENCE OF POSITIONS

We introduce the U-shaped Placement strategy to organize document positions in accordance with the model's positional bias. This method is compatible with any document relevance ordering, regardless of the ranking method employed. In this section, we utilize an external relevance ranking and focus on assessing the effect of document placement.

We evaluate four placement strategies: placing relevant documents near the question, U-shaped Placement, and the reverse variants of both. In the original versions, higher-relevance documents are assigned to positions that inherently receive more attention, while the reverse versions deliberately assign lower-relevance documents to these favored positions. All four configurations use identical prompt templates, varying only in document order, thereby isolating the effect of placement on model performance. We've included an example in the Appendix E for better explanation.

Table 3: Results of different placements with relevance ranking based on external retrieval. Default means placing the relevant ones close to question, while U-shaped is our method in accordance with positional bias. Different settings indicate whether a good position is preferentially occupied by a good (original) or bad (reverse) document.

| Placement | Setting | Vicuna-7b | | | Llama-3.1-8b | | | Qwen2.5-7b | | | Qwen2.5-7b-ins | | |
|---|---|---|---|---|---|---|---|---|---|---|---|---|---|
| | | H | M | W | H | M | W | H | M | W | H | M | W |
| Default | Original | 0.4 | 0.312 | 0.482 | 0.302 | 0.238 | 0.358 | 0.408 | 0.342 | 0.41 | 0.472 | 0.5 | 0.526 |
| | Reverse | 0.378 | 0.24 | 0.464 | 0.28 | 0.2 | 0.348 | 0.37 | 0.323 | 0.39 | 0.456 | 0.477 | 0.5 |
| U-shaped (Our) | Original | 0.397 | 0.314 | 0.524 | 0.31 | 0.252 | 0.376 | 0.416 | 0.369 | 0.414 | 0.486 | 0.509 | 0.522 |
| | Reverse | 0.367 | 0.216 | 0.446 | 0.268 | 0.186 | 0.338 | 0.368 | 0.3 | 0.39 | 0.454 | 0.435 | 0.476 |

The results in Table 3 demonstrate that: (1) The proposed U-shaped Placement, which aligns with the model's positional bias, represents a more effective placement strategy that enhances the model's ability to utilize documents. When compared with the results in Table 2, the performance of Vicuna-7b and Llama-3.1-8b models approaches or even exceeds that in the main experiments, whereas the Qwen series still lags behind. This observation aligns with earlier findings regarding model capabilities, suggesting that relevance rankings produced by the Qwen series are comparatively more reliable. (2) Among the reverse placement configurations, the reversed U-shaped Placement leads to the most significant performance degradation, underscoring the importance of leveraging positional bias.

## 6.2 DOCUMENT RELEVANCE

In prior analyses, we aggregated attention weights from context tokens to answer tokens to estimate document importance. In this section, we investigate the effect of different document relevance sorting methods and aggregation strategies, mainly considering aggregation based on the first generated token or query tokens involved in the previous works.

Table 4: Results of the different document relevance sorting methods of vicuna-7b model.

| Level | Source | H | M | W |
|---|---|---|---|---|
| Document-Level | Retrieval | 0.4 | 0.312 | 0.482 |
| | RankGPT | 0.403 | 0.28 | 0.502 |
| Token-level | First Token | 0.404 | 0.291 | 0.48 |
| | Query | 0.398 | 0.301 | 0.49 |
| | Answer | 0.406 | 0.305 | 0.494 |
| | Answer(qwen) | 0.412 | 0.329 | 0.512 |

To exclude positional effect, we adopt the default ranking strategy, which places the most relevant documents closest to the question. Results of vicuna-7b in Table 4 show that: (1) Aggregation based on answer tokens outperforms document-level, query-based and first-token-based approaches, as it more directly captures influence on the generated answer. (2) As mentioned before, document relevance rankings derived from Qwen models are more reliable. Providing such rankings to Vicuna improves its performance, suggesting a promising direction for hybrid approaches that leverage multiple models during generation.

## 6.3 THE NUMBER OF DOCUMENTS

While the main experiments validate the effectiveness of our method on datasets containing 10 and 20 documents, we also conducted additional experiments with fewer input documents to assess its versatility and efficacy. We take the original ten-document 2wikiMultiHopQA dataset and only intercepted the first three and five documents for the experiment. Experimental results in the Table 5 demonstrate that our proposed ranking method and U-shaped Placement remain high efficiency across varying numbers of documents.

Table 5: Results of varying number of input documents.

| Prompt | Methods | Vicuna-7b | | | Qwen2.5-7b-ins | | |
|---|---|---|---|---|---|---|---|
| | | 3doc | 5doc | 10doc | 3doc | 5doc | 10doc |
| Unordered | Vanilla + Default | 0.286 | 0.296 | 0.452 | 0.316 | 0.334 | 0.48 |
| | Vanilla + U-shaped | 0.334 | 0.35 | 0.48 | 0.34 | 0.358 | 0.501 |
| | Our Ranking + Default | 0.344 | 0.358 | 0.474 | 0.342 | 0.35 | 0.51 |
| | Our Ranking + U-shaped | **0.356** | **0.374** | **0.506** | **0.35** | **0.36** | **0.536** |
| Ordered | Vanilla + Default | 0.38 | 0.424 | 0.482 | 0.366 | 0.428 | 0.526 |
| | Vanilla + U-shaped | 0.398 | **0.47** | **0.524** | 0.37 | 0.464 | 0.522 |
| | Our Ranking + Default | 0.39 | 0.456 | 0.494 | 0.376 | 0.444 | 0.53 |
| | Our Ranking + U-shaped | **0.401** | 0.463 | 0.51 | **0.394** | **0.466** | **0.542** |

## 7 CONCLUSION

In this paper, we investigate the positional bias based on model's attention weight, both horizontally and vertically. We find that the model's estimation of document importance is also internally affected by positional bias in a U-shape, with the magnitude of the U-shape varying with the order of input documents. In addition, the lower layers reflect the position information more significantly.

And we propose U-shaped Placement to separate and utilize positional bias. Combining it with the importance estimation of documents within the model, placing good documents in good positions, can improve the model's ability to utilize documents within two iterations. Our approach requires no training, and can work on any open-source model and dataset.

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

## A    LLM Usage

Large Language Models (LLMs) are increasingly utilized in scientific research and provide substantial support in academic writing. Their applications range from enhancing grammar and wording to assisting in the drafting of complete manuscript sections. In this paper, we employed an LLM solely for language refinement aimed at improving clarity and explanatory quality. All content has been thoroughly verified for factual accuracy, and the authors take full responsibility for the entirety of the work. The central ideas, experimental design, and methodological framework were developed independently by the authors without the use of LLMs.

## B    Details about Preliminary Experiments

### B.1    Datasets

We applied the datasets processed by Pan et al. (2024) in our paper. Due to resource limitations, we mainly focus on several open-domain variants of the datasets.

HotpotQA (Yang et al., 2018) and 2WikiMHQA (Ho et al., 2020) both require reasoning across multiple documents, and feature a high proportion of distracting documents. Importantly, the data from HotpotQA is extracted from the dev subset, whereas our training dataset is derived from the train subset. Musique (Trivedi et al., 2022) questions are of higher complexity, with up to 90% of distracting passages.

See original paper (Pan et al., 2024) for more details.

## B.2 IMPLEMENTATION DETAILS

We will list the details of hyperparameters we used in the experiments. The seed is set to 42. The temperate is set to 0.01 and the number of max_new_tokens is 300. The same prompt template is used for all datasets and all models in the experiments to exclude template interference, which is presented as follows:

> You're a helpful AI assistant. The assistant answers questions based on given passages.
>
> Docs:{{$d_0$.title}}:{{$d_0$.text}}
> {{$d_1$.title}}:{{$d_1$.text}}
> {{$d_2$.title}}:{{$d_2$.text}}
>
> (more passages) ...
>
> Question: {{question}}
>
> Answer:

## C MIRAGE RESULTS

An ordered placement approach such as placing relevant documents close to the questions is a powerful baseline, but we want to make better use of the model's positional bias. Therefore, we first explore the dependency of the generated answer token on the context documents using the library developed by Qi et al. (2024). MIRAGE identifies context-sensitive answer tokens and aligns them with retrieved documents based on internal model states. We further analyze the positional distribution of context documents that answer tokens attend to most.

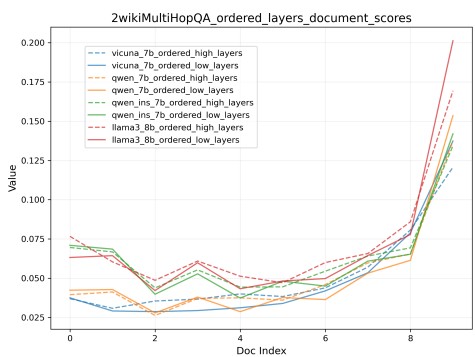

Figure 4: The document scores $S$ of all models with different selected layers on 2wikiMulti-HopQA datasets under ordered input.

The results of vicuna-7b, llama3-8b, qwen-7b, and qwen-7b-instruct are presented in Figure 5a, 5b, 5c, and 5d, respectively.

The different rows represent the results on different datasets: HotpotQA, Musique, 2wikiMulti-HopQA. The different columns represent the different ways of composing the prompt: unordered (concat) or ordered (rerank). In each figure, the horizontal axis represents document positions within the prompt, ranging from position 0 (beginning) to position $N-1$ (end, closest to the question). The vertical bar indicates the number of answer tokens that depend on the document located at each corresponding position.

The results show that under ordered input, it is common sense to depend on the documents near the question. In contrast, it shows a clear positional bias towards the beginning and the end under unordered input, which matches the Lost-in-the-middle (Liu et al., 2024) phenomenon in performance. And this is more evident on the Musique which has a larger number of documents.

## D ATTENTION WEIGHT RESULTS

The complete Attention Weight Results are presented here.

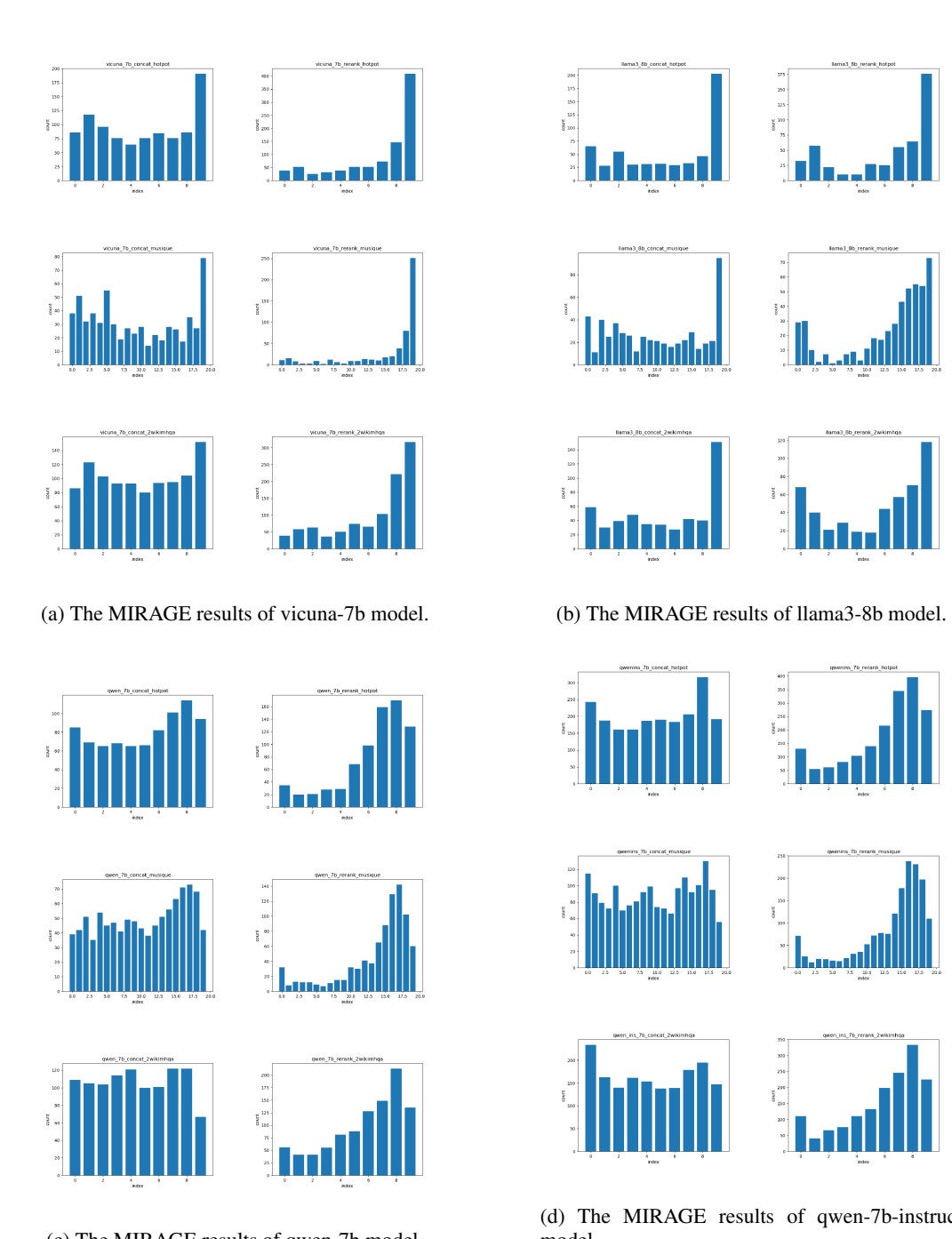

(a) The MIRAGE results of vicuna-7b model.

(b) The MIRAGE results of llama3-8b model.

(c) The MIRAGE results of qwen-7b model.

(d) The MIRAGE results of qwen-7b-instruct model.

Figure 5: The MIRAGE results of all models. For each model, the different lines represent different datasets: HotpotQA, Musique, 2wikiMultiHopQA. The first and second columns represent the unordered and ordered inputs.

## D.1 HORIZONTAL RESULTS

We present the complete results of the difference between the score of different documents in different order of documents in this section. The results on HotpotQA dataset is presented in Figure 6a, and the results on Musique dataset is presented in Figure 6b.

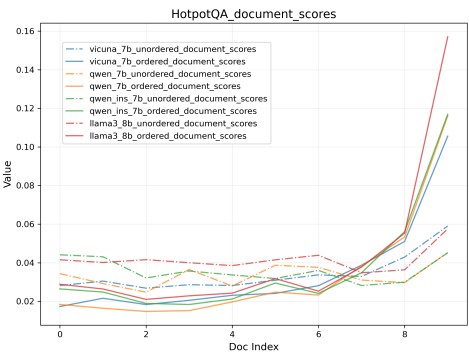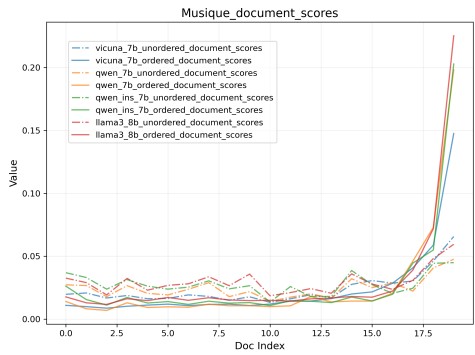

(a) The document scores $S$ of all models on HotpotQA datasets.

(b) The document scores $S$ of all models on Musique datasets.

Figure 6: The document scores $S$ of all models on HotpotQA datasets. The solid line - corresponds to the ordered input, the dashed -. line corresponds to the unordered input, and the results of the same model are shown in the same color.

## D.2 VERTICAL RESULTS

We present the complete results of different selected layers in this section. See Figure 4,7a, 7b,7c,7d for more information.

## D.3 POSITIONAL SCORES

We present the complete results of postional scores on all datasets in this section. See Figure 8a,8b for more information.

## E EXAMPLES OF DIFFERENT ORDERING

The goal of original ranking and U-shaped Placement is to place good documents in good positions, but the default good positions are different. As an example, if the dataset has 10 documents, the order of documents under the ordered input is [0,1,.... ,9], the question is placed at the end, document 9 has the best relevance, and document 0 has the worst relevance. After placing the documents according to the positional bias under the U-shaped Placement, the order of documents may become [6,5,4,2,0,1,3,7,8,9], and the question is placed at the end as well. While the reverse version of ordered has the input document order as [9,8,... ,0], and the reverse version of U-shaped Placement has the document order [3,4,5,7,9,8,6,2,1,0], with bad documents prioritized to occupy the default good placements in each reverse order.

## F RANKGPT

The prompt template used during the first round of RankGPT generation is as follows, based on which the prompts are constructed to allow LLM to perform listwise document sorting.

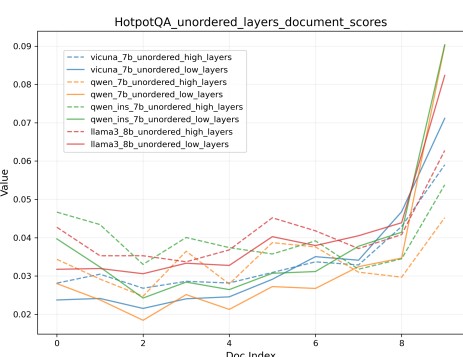

(a) The document scores $S$ of all models with different selected layers on Hotpot datasets under unordered input.

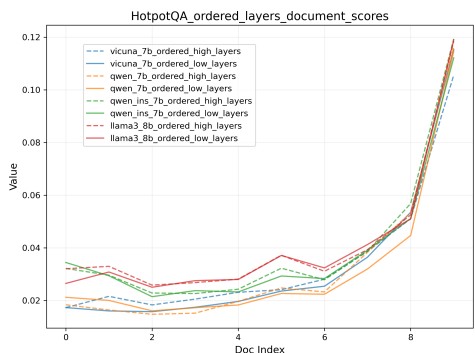

(b) The document scores $S$ of all models with different selected layers on Hotpot datasets under ordered input.

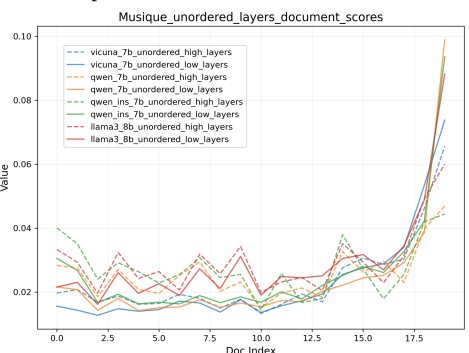

(c) The document scores $S$ of all models with different selected layers on Musique datasets under unordered input.

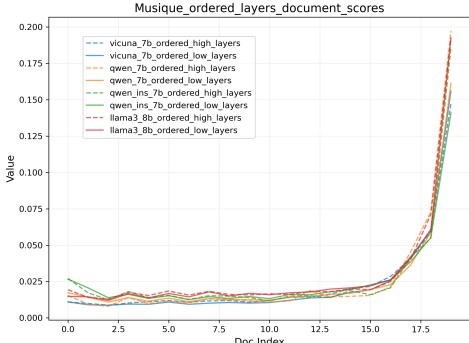

(d) The document scores $S$ of all models with different selected layers on Musique datasets under ordered input.

Figure 7: The document scores $S$ of all models with different selected layers. The solid - and dotted − lines are used to distinguish the first and the last half of layers. And the results of the same model are shown in the same color.

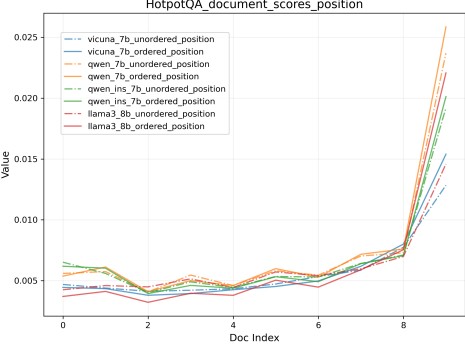

(a) The positional scores $S$ of all models on HotpotQA datasets.

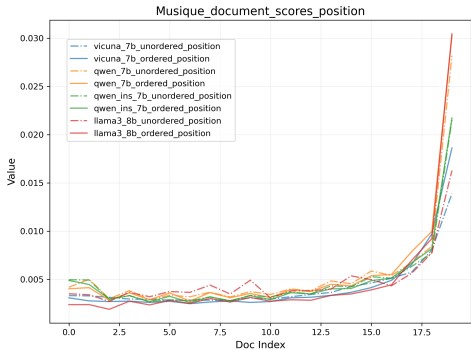

(b) The positional scores $S$ of all models on Musique datasets

Figure 8: The positional scores $S$ of all models, which are calculated by aggregating the document scores of the previous token and the terminating token of the answer tokens.

Table 6: Results of the different document relevance sorting methods of vicuna-7b model under ordered input. Calibration means subtracting positional influence from attention scores.

| Ranking | Aggregation | H | M | W |
|---|---|---|---|---|
| Retrieval | - | 0.4 | 0.312 | 0.482 |
| Attention Weight | answer | 0.406 | 0.305 | 0.494 |
|  | calibration | 0.398 | 0.279 | 0.496 |

> This is RankGPT, an intelligent assistant that can rank passages based on their relevancy to the query.
>
> The following are {{num}} passages, each indicated by number identifier []. I can rank them based on their relevance to query: {{query}}
>
> [1] {{passage_1}}
>
> [2] {{passage_2}}
>
> (more passages) ...
>
> The search query is: {{query}}
>
> I will rank the {{num}} passages above based on their relevance to the search query. The passages will be listed in descending order using identifiers, and the most relevant passages should be listed first, and the output format should be [] ¿ [] ¿ etc, e.g., [1] ¿ [2] ¿ etc.
>
> The ranking results of the {{num}} passages (only identifiers) is:

## G  SOMETHING WE TRIED

The complete pipeline of our proposed algorithm is embodied in Algorithm 2, while in this section we briefly describe some additional attempts at details.

First, we address the estimation of document relevance. In previous experiments, we directly utilized document scores as the basis for estimation. The calibration method introduced by Chen et al. (2024) offers a valuable inspiration. Accordingly, we also attempt to remove positional effects from the attention scores. Specifically, we subtract the positional scores from the document relevance scores. However, this approach yields no significant improvement, likely because the answer-based aggregation scores and the positional representations (derived from start and end tokens) are not strictly commensurable. The corresponding results are provided in G.1.

We place the documents according to the U-shape in our proposed method, however, the positional bias does not exactly fit the U-shape and there may be zigzag in the middle, as shown in previous analysis. Aggregating the token-level position scores by document and then placing the document directly according to the result of document-level has no zigzag problem, but it has length problem as said in section 4. Is the length issue more important or the zigzag issue? The results in G.2 show that placement according to the U-shape is more in line with the positional bias, and the length mismatch has a greater impact on performance compared to the zigzag problem.

### G.1  CALIBRATION

As in section 6.2, the vicuna model was also used in the experiments under the ordered input and the results are presented in Table 6.

Table 7: Results of different placements after sorting them for relevance based on external search scores. Default means directly placing the relevant ones close to the questions, while U-shaped is our proposed method in accordance with positional bias. Direct-U means aggregating token-level position scores by document and then placing the document directly according to the result of document-level.

| Placement | Vicuna-7b | | | Llama-3.1-8b | | | Qwen2.5-7b | | | Qwen2.5-7b-ins | | |
|---|---|---|---|---|---|---|---|---|---|---|---|---|
| | H | M | W | H | M | W | H | M | W | H | M | W |
| Default | 0.4 | 0.312 | 0.482 | 0.302 | 0.238 | 0.358 | 0.408 | 0.342 | 0.41 | 0.472 | 0.5 | 0.526 |
| U-shaped (Our) | 0.397 | 0.314 | 0.524 | 0.31 | 0.252 | 0.376 | 0.416 | 0.369 | 0.414 | 0.486 | 0.509 | 0.522 |
| Direct-U | 0.39 | 0.291 | 0.49 | 0.31 | 0.232 | 0.356 | 0.406 | 0.361 | 0.406 | 0.472 | 0.495 | 0.516 |

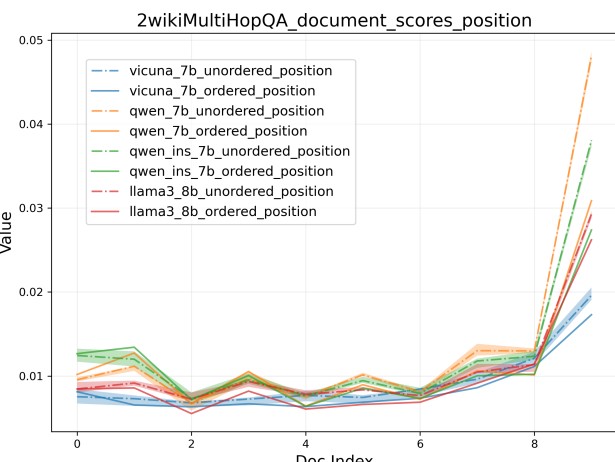

Figure 9: The results from five repeated experiments.The shaded area indicates the range of score variations across these five random experiments.

### G.2 ZIGZAG

The results are presented in Table 7.

## H  RANDOM VARIATION

To eliminate the impact of random variation while enhancing the credibility of our conclusions, we conduct repeated experiments using different random number seeds for experiments in Figure 3. We plotted the range of variation in the results from five repeated experiments as shaded areas in the figure 9 .

The results demonstrate that even with multiple randomizations of document order, the calculated outcomes exhibit a certain degree of stability.

