# OpenReview forum: "From Bias to Benefit: Place Good Documents in Good Positions"
_ICLR.cc/2026/Conference — ICLR 2026 Conference Withdrawn Submission_

### Official Review · Reviewer_nuGU · 2025-10-17

**Soundness:** 3
**Presentation:** 3
**Contribution:** 2
**Rating:** 4
**Confidence:** 4

**Summary:**

This paper proposes a 2-stage method that puts more important documents on the beginning or end of the input. The motivation is to utilize the U-shaped position bias of LLMs to make the more important documents get more attention.
In stage 1, to rank the importance of each document, it uses averaged attention weights from answer tokens. Moreover, it separates a "position bias score" to decide whether a document should be placed at the beginning or end of the input. In stage 2, LLM uses the reordered input to generate the answer.
Experiments on 3

**Strengths:**

1. Clear structure and easy to understand
2. Experiments on many baseline methods and datasets

**Weaknesses:**

1. This method is similar to baseline methods such as Attention Sorting, so the novelty is limited. The only difference is to put the document either on the beginning or the end based on a "position bias score".
2. The findings and insights are trivial for me. Most are already found by previous works about lost-in-the-middle, but these works are not cited in Related works. For example:

[1] Found in the Middle: Calibrating Positional Attention Bias Improves Long Context Utilization

[2] Found in the Middle: Permutation Self-Consistency Improves Listwise Ranking in Large Language Models

[3] Eliminating Position Bias of Language Models: A Mechanistic Approach

[4] Mitigate Position Bias in Large Language Models via Scaling a Single Dimension

3. This method relies on attention weights. However, calculating attention weights means more computing, and the inability to use FlashAttention2 in generation. This will significantly slow down the inference speed, especially when the input is very long. So it is not so practical.
4. The improvement compared to baselines is not so great.

**Questions:**

none

---

> ### Author Response · Authors · 2025-11-21
> **Response to Reviewer nuGU: Weakness**
>
> Thank you for your valuable feedback and constructive suggestions. We are committed to addressing these points to improve the quality of the paper.
>
> **R W1: Novelty of Our Work**
>
> Thanks for your question. In this paper, we propose separating positional bias from attention weights and leveraging it. This is a dynamic process that requires neither training nor pre-specified hyperparameters.  Our proposed placement method can be combined with any relevance estimation method and achieves better results than sequential placement. Furthermore, our experiments demonstrate that going against positional bias significantly impacts performance, further highlighting the importance of utilizing positional bias.
>
> **R W2：Lack of Related Works**
>
> Thank you for bringing this to our attention. We did indeed overlook certain aspects during our research process. We will enhance the “Related Work” section to incorporate these studies.
>
> At the same time, we contrast our work with these studies to further highlight our contributions.
>
> - [1] ：This paper also examines the impact of position from the perspective of attention weight. However, we introduce at least three distinct aspects: (1) They employed TF-IDF to measure the dependence of answers on documents. However, TF-IDF fails to indicate whether LLMs genuinely relied on the document to generate the answer. We adopted the MIRAGE metric, which better aligns with the generative characteristics of LLMs.（2）They only studied it from a horizontal perspective, while we also examined the vertical（layers） impacts.（3）Most importantly, they propose that positional bias and relevance form an additive relationship, which was not reflected in our experiments (see Appendix G and table 6).We believe the relationship is likely more complex, and our approach can leverage positional bias even when the exact combination is unknown.
> - [2]：Their key idea is to marginalize out different list orders in the prompt to produce an order-independent ranking with less positional bias, which consumes more computational resources during inference. Our approach involves only two rounds of iteration.
> - [3],[4]：Similarly, they explore the causes of position bias and propose measures to mitigate its effects. Both papers are highly insightful. However, the proposed solutions involve direct modifications to attention or positional encoding mechanisms, requiring training to ensure optimal performance. Our approach delivers improvements without requiring training.
>
> Their objectives aim to eliminate the influence of positional bias, whereas our approach leverages its effects. Even when the precise mathematical relationship between positional bias and document relevance within attention weights remains unknown, our method achieves improvements without requiring additional training.
>
> References:
>
> [1] Found in the Middle: Calibrating Positional Attention Bias Improves Long Context Utilization
>
> [2] Found in the Middle: Permutation Self-Consistency Improves Listwise Ranking in Large Language Models
>
> [3] Eliminating Position Bias of Language Models: A Mechanistic Approach
>
> [4] Mitigate Position Bias in Large Language Models via Scaling a Single Dimension
>
> **R W3: Inability to Use FlashAttention2**
>
> Thanks for your question. Our approach relies on attention weights, requiring the obtain of attention weight values during generation and subsequent computation based on these values. While this does increase computational overhead during inference, it remains significantly more efficient than methods requiring training. And our work aims to advance academic exploration rather than pursue industrial applications.
>
> **R W4: Improvement**
>
> Thanks for your question. Our approach outperforms various baselines both under ordered inputs and the more challenging unordered inputs. Compared to traditional RAG workflows, it delivers significant improvements, particularly on datasets with a large number of documents, achieving a 5-10 percentage point performance boost.

---

### Official Review · Reviewer_tut1 · 2025-10-29

**Soundness:** 3
**Presentation:** 2
**Contribution:** 2
**Rating:** 4
**Confidence:** 3

**Summary:**

This paper investigates the U-shaped position bias in LLM-based RAG systems. The authors propose a new method to isolate position bias in LLMs and design a U-shaped placement strategy to organize input documents. Extensive experiments are conducted to validate the effectiveness of the proposed approach.

**Strengths:**

1.The paper provides a thorough and detailed study of the U-shaped position bias phenomenon.

2.The related work is well-summarized, and the connection to prior research is clearly articulated.

3.The proposed solution is simple and efficient.

**Weaknesses:**

1.In Step 1, the authors use attention weights to estimate document importance. However, since attention weights themselves may be influenced by position bias, it is unclear whether same documents placed in different positions would yield consistent importance scores. This potential confounding effect needs to be clarified and ideally supported with empirical evidence.

2.In Lines 193–195, when discussing the relationship between position bias and the number of documents, the authors should provide the exact document counts for each dataset. Moreover, this relationship should also be tested within the same dataset by varying the number of documents to ensure a fair comparison.

3.Regarding the finding in Lines 257–259—“positional distinctions are more pronounced in the lower layers”—this may not hold for ordered inputs. As shown in Appendix D.2, the curves for lower and higher layers appear nearly identical under ordered settings, suggesting that the distinction is much less evident.

4.In addition to Figure 3, the accuracy and stability of the calculated position biases should be further justified. For instance, when document orders are randomized multiple times, do the computed biases remain consistent? Providing more quantitative evidence and analysis of variance across random orders would strengthen this point.

5.The experiments are limited to small-scale LLMs. It would strengthen the paper to evaluate the method’s scalability on larger models (e.g., 10B or 30B parameters) to confirm its scalability.

**Questions:**

1.Several figures (e.g., Figures 2 and 3) are of low resolution. They should be redrawn using vector formats such as .eps or .pdf. Additionally, increasing line weight and font size would improve readability.

2.There are some typos in the paper, e.g., line 152 Dlanguage and line 758 lowercase "we".

---

> ### Author Response · Authors · 2025-11-21
> **Response to Reviewer tut1: Question and Weakness (Part I)**
>
> Thank you for your valuable feedback and constructive suggestions. We are committed to addressing these points to improve the quality of the paper.
>
> **R W1: Attention weights themselves may be influenced by position bias**
>
> Thanks for your question. Your concerns are also ours, so we conducted experiments in the appendix to attempt to obtain document relevance scores unaffected by positional bias (see Appendix G and table 6). However, this substration approach yields no significant improvement. There may be several possible reasons for this phenomenon and your question:
> - **The combination of positional bias and document relevance scores in attention weights is not a simple additive relationship, but rather a more complex combinatorial relationship.**  Therefore, our approach of subtracting position scores before reordering did not yield significant improvements. Our method can leverage positional bias without knowing the specific combination relationship.
> - **Although the specific combination relationship remains unknown, the attention weight exhibits a monotonically increasing relationship with document relevance.** If a document's attention weight is low, then it means that its semantic relevance score cannot be high. And the attention score is more influenced by semantic relevance, and positional bias cannot play a decisive role. This is exemplified by the difference in the maximum value of the vertical axis in Fig. 2 and Fig. 3. Because the values of positional bias score are not high, we propose to place them according to the U-shape instead of adding or subtracting them directly.
> - **The placement method we propose can be combined with any relevance estimation methods.** When combined with external retrieval ranking, it also achieves excellent results.
>
> We will continue to investigate the confounding effect between positional bias and document relevance.
>
> **R W2: Number of Documents**
>
> Thanks for your question. In the experiments conducted in this paper, we utilized the dataset processed by previous work, which contains 10 documents for HotpotQA, 20 documents for Musique, and 10 documents for 2WikiMHQA. Indeed, what we aim to illustrate in lines 193–195 is that the effect of positions becomes more pronounced in datasets with a larger number of documents. We will modify the expression to avoid ambiguity.
>
> Regarding the fairness issue, we have re-obtained the results by adjusting the number of documents in the same dataset as you suggested. We conducted experiments on the Musique dataset using different numbers of documents (10, 20), with results shown in the table. As mentioned in the paper, as the number of documents increases, the effect of position becomes more pronounced.
>
> | Model       | Setting | M-10doc | M-20doc |
> |-------------|--------|---------|---------|
> | vicuna-7b   | unordered | 0.228   | 0.238   |
> |             | ordered   | 0.256   | 0.312   |
> | Llama-3.1-8b| unordered | 0.174   | 0.192   |
> |             | ordered   | 0.194   | 0.238   |
> | Qwen2.5-7b| unordered | 0.27   | 0.298   |
> |             | ordered   | 0.302  | 0.342  |
> | Qwen2.5-7b-ins| unordered | 0.37   | 0.458   |
> |             | ordered   | 0.406  | 0.5  |
>
>
> **R W3：The distinction is much less evident under ordered input.**
>
> Thanks for your question. Under ordered input, the differences between layers are indeed less pronounced than under unordered input. It may be because, under these circumstances, the relevance score dominates the attention weight. However, the position scores we separated in Figure 4 still exhibit a U-shaped pattern.  To better demonstrate that lower layers are more suitable than higher layers for detecting positional information, we present additional experimental results. We use the retrieval ranking as the document order as table 3 and evaluate the performance of placement when selecting low layers and high layers.
>
> | Model       | Setting | H | M| W |
> |-------------|--------|---------|---------|---------|
> | vicuna-7b | Low Layers| 0.397 | 0.314 | 0.524 |
> |           | High Layers | 0.388 | 0.306 | 0.498 |
> | llama-3.1-8b | Low Layers| 0.31 | 0.252 | 0.376 |
> |           | High Layers | 0.302 | 0.222 | 0.354 |
> | Qwen2.5-7b | Low Layers| 0.416 | 0.369 | 0.414 |
> |           | High Layers | 0.394 | 0.364 | 0.382 |
> | Qwen2.5-7b-ins | Low Layers| 0.486 | 0.509 | 0.522 |
> |           | High Layers | 0.474 | 0.494 | 0.51 |
>
> **The results indicate that positional information obtained from lower layers is more helpful for document arrangement.** We hope this will address your concerns from another perspective.

---

> > ### Author Response · Authors · 2025-11-21
> > **Response to Reviewer tut1: Question and Weakness (Part II)**
> >
> > **R W4：When document orders are randomized multiple times, do the computed biases remain consistent?**
> >
> > Thanks for your questions. The dataset used in our experiment was preprocessed from previous work, and a randomized version was also provided rather than being processed by us. To better address your concerns about random variation while enhancing the credibility of our conclusions, we conducted repeated experiments using different random number seeds. We plotted the range of variation in the results from five repeated experiments as shaded areas in the figure. Since images cannot be directly uploaded during the rebuttal process, we have included the results in **Appendix H** of the updated paper. **The results demonstrate that even with multiple randomizations of document order, the calculated outcomes exhibit a certain degree of stability.**
> >
> > **R W5: Limited to Smaller-scale Models**
> >
> > Thanks to your question. Although computational resource constraints prevented us from testing on larger models (70B), we conducted additional experiments on a 13B model to demonstrate the generalization capability of our approach.
> >
> > |vicuna-13b|H-unordered|M-unordered|W-unordered|H-ordered|M-ordered|W-ordered|
> > |---|---|---|---|---|---|---|
> > |vanilla|0.386|0.264|0.442|0.406|0.313|0.482|
> > |RankGPT|0.388|0.303|0.438|0.408|0.301|0.456|
> > |AttentionSort|0.41|0.305|0.458|0.408|0.301|0.474|
> > |ICR|0.406|**0.313**|0.479|0.405|0.343|0.475|
> > |SELFELICIT|0.382|0.294|0.447|0.402|0.334|0.478|
> > |Our|**0.42**|0.311|**0.484**|**0.414**|**0.357**|**0.497**|
> >
> > The results are similar to those in Table 1, demonstrating that **our method outperforms the previous baseline on the 13B model** and enhances the model's ability to utilize documentation.
> >
> > **R Q: Figure and Typos**
> >
> > We apologize for these minor typos. We have uploaded a PDF version with the figures replaced as you suggested and corrected the typos.

---

### Official Review · Reviewer_fxGj · 2025-10-30

**Soundness:** 2
**Presentation:** 2
**Contribution:** 2
**Rating:** 4
**Confidence:** 4

**Summary:**

This paper investigates the "Lost in the Middle" phenomenon—where models tend to assign higher attention to the beginning and end of input prompts while neglecting the middle—from the perspective of the model's attention mechanism. It constructs a position score by aggregating attention weights before and after answer tokens. Based on document importance scores and position scores, the paper proposes a strategy called U-shaped Placement, which rearranges documents to align with the model’s inherent positional bias, ensuring that highly relevant content is placed in positions that receive greater attention.

**Strengths:**

1. The investigation of the "Lost in the Middle" phenomenon from the perspective of the attention mechanism may better reveal its underlying causes.

2. Based on the insights gained from this investigation, the proposed U-shaped Placement strategy is supported by comprehensive experiments.

**Weaknesses:**

1. There is a lack of ablation studies on the token-level scores. A more rigorous validation of the rationale behind using token-level (as opposed to document-level) scores would involve replacing only the scoring granularity while keeping all other factors constant.

2. The experiments are limited to smaller-scale models, leaving it unclear whether the U-shaped Placement strategy generalizes to larger LLMs with more parameters.

3. The paper's exposition could be clearer. The distinction between related prior work and the novel contributions of this work is not sufficiently emphasized.

**Questions:**

1. Why do the results for "U-shaped Original" in Table 3 differ from those in Table 2? Are the results in Table 3 obtained by replacing the token-level score with a document-level score? If so, what distinguishes this document-level version from existing methods?

2. What is the specific difference between the token-level method proposed in this paper and the token-level method(s) mentioned in Table 4?

​Comment:​​

If the authors can clarify the distinctions between their designed method and existing approaches, thereby providing a clearer understanding of its advantages, I would consider raising my score.

---

> ### Author Response · Authors · 2025-11-21
> **Response to Reviewer fxGj: Questions**
>
> Thank you for your valuable feedback and constructive suggestions. We are committed to addressing these points to improve the quality of the paper.
>
> **First: Contribution and Distinction**
>
> First, it seems you may be confused about the distinction between our work and previous works, so we first explain our distinction and contribution. We first examined positional bias from both horizontal and vertical perspectives, then proposed a pipeline to leverage positional bias. Our proposed pipeline consists of two components:
>
> - **U-shaped Placement. This is also the most significant difference and contribution of this paper compared to previous work.** We propose using aggregations over meaningless tokens to obtain position scores, and arranging documents in a U-shaped pattern based on these scores to ensure relevant documents receive greater attention.
> - **Document ranking based on attention weights aggregated from context tokens to answer tokens.** We believe that scores obtained by aggregating answer tokens more intuitively reflect the influence of context on answers.
>
> To better illustrate our findings, we present a tabular comparison of our work alongside previous related studies:
>
> | | Document relevance estimation method| Placement stratety |
> |---|---|---|
> |vanilla|Retrieval ranking|sequential placement|
> |Attention Sorting|Aggregate the attention weights from the context tokens to the **first token** as the basis for ranking.|sequential placement|
> |ICR| Aggregate the attention weights from the context tokens to the **query tokens** and employ meaningless query for calibration.|sequential placement|
> |SELFELICIT|Aggregate the attention weights from context tokens to **first tokens** as the basis for determining sentence importance.|sequential placement. Mark important sentences|
> |Our| Aggregate the attention weights from the context tokens to the **answer tokens** as the basis for ranking| **U-shaped placement**, arranging documents in a U-shaped pattern based on seperated positional scores.|
>
> In Table 2, we demonstrate that the framework composed of these two components outperforms the previous two rounds of iterative work. Tables 3 and 4 **respectively** highlight the importance of U-shaped Placement and answer token aggregation strategy.
>
> **R Q1: Why do the results for "U-shaped Original" in Table 3 differ from those in Table 2?**
>
> Thanks for your question. In this paper, we propose a placement method called U-shaped Placement that leverages positional bias. The method requires inputting the order of document relevance and then placing relevant documents in positions where positional bias can be leveraged. Therefore, this placement method can be combined with different relevance estimation methods. In Table 2, ranking documents based on aggregated attention scores is the relevance estimation method we employed, while retrieval ranking served as the relevance estimation method utilized in Table 3. **And different relevance estimation methods will yield different results. And the most significant difference between our approach and existing methods lies in the placement of documents.** While existing methods arrange documents sequentially, we dynamically utilize positional bias along a U-shaped curve to position documents. The results in Tables 2 and 3 both demonstrate the superiority of our placement method.
>
> **R Q2：What is the specific difference between the token-level method proposed in this paper and the token-level method(s) mentioned in Table 4?**
>
> Thanks for your question. In Table 4, we did not employ the proposed U-shaped placement but instead used the default sequential placement to compare the effects of sorting based on different methods. The aggregation method we propose for document ranking aggregates the attention weights from context tokens to answer tokens. Previous token-level approaches employed different aggregation scopes, such as aggregating context tokens to query tokens or to the first token. Our experimental results in Table 4 demonstrate that **aggregating to the answer token yields superior ranking performance.** When combined with our proposed U-shaped Placement, placing good documents in good positions, it achieves the best results (see Table 1).

---

> > ### Author Response · Authors · 2025-11-21
> > **Response to Reviewer fxGj: Weakness**
> >
> > **R W1: Lack of Ablation Studies on the Token-level Scores.**
> >
> > I'm not sure if I've fully understood your question. Please correct me if I have misinterpreted it, and I would be happy to provide further clarification.
> > Our scores are obtained based on token-level attention weight. **The experiments in Tables 1 and 2 in the paper both involve comparisons with document-level scores.**  Among the baselines we compare, Vanilla uses the document-level score given by an external retriever, and RankGPT uses the document-level score given by the model itself through prompting. The results in Table 1 are inclusive of positional processing, whereas the results in Table 2 are comparisons using token-level and document-level score sorting. For better demonstration, we extract and redesign the results as follows:
> >
> > | method                | granularity     | vicuna-7b         | Qwen2.5-7b          |
> > |-----------------------|-----------------|--------------------|---------------------|
> > | Vanilla               | document-level  | 0.4/0.312/0.482    | 0.408/0.342/0.41    |
> > | RankGPT               | document-level  | 0.403/0.28/0.502   | 0.391/0.358/0.416   |
> > | Our attention weight  | token-level     | 0.406/0.305/0.494  | 0.41/0.369/0.42     |
> > | Our attention weight + U | token-level     | **0.426/0.315/0.51** | **0.412/0.401/0.436** |
> >
> > The three numbers in the table are the results of HotpotQA, Musique, and 2wikiMultiHopQA, respectively, which show that **token-level scores outperform document-level**, and that further use of U-shaped Placement based on sorting performs the best.
> >
> > We are not the first to use token-level attention scores as the basis for document arrangement. Previous works[1][2][3] also use this design and clarify that token-level scores are indeed superior to document-level. Our key design is the U-shaped Placement that places relevant documents according to the positional bias.
> >
> > References:
> >
> > [1] ATTENTION SORTING COMBATS RECENCY BIAS IN LONG  CONTEXT LANGUAGE MODELS
> >
> > [2] Attention in Large Language Models Yields Efficient Zero-Shot Re-Rankers
> >
> > [3] SelfElicit: Your Language Model Secretly Knows Where is the Relevant Evidence
> >
> > **R W2: Limited to Smaller-scale Models**
> >
> > Thanks for your question. Although computational resource constraints prevented us from testing on larger models (70B), we conducted additional experiments on a 13B model to demonstrate the generalization capability of our approach.
> >
> > |vicuna-13b|H-unordered|M-unordered|W-unordered|H-ordered|M-ordered|W-ordered|
> > |---|---|---|---|---|---|---|
> > |vanilla|0.386|0.264|0.442|0.406|0.313|0.482|
> > |RankGPT|0.388|0.303|0.438|0.408|0.301|0.456|
> > |AttentionSort|0.41|0.305|0.458|0.408|0.301|0.474|
> > |ICR|0.406|**0.313**|0.479|0.405|0.343|0.475|
> > |SELFELICIT|0.382|0.294|0.447|0.402|0.334|0.478|
> > |Our|**0.42**|0.311|**0.484**|**0.414**|**0.357**|**0.497**|
> >
> > The results are similar to those in Table 1, demonstrating that **our method outperforms the previous baseline on the 13B model** and enhances the model's ability to utilize documentation.
> >
> > **R W3: Contribution and Distinction**
> >
> > See *First: Contribution and Distinction*

---

> > > ### Comment · Reviewer_fxGj · 2025-11-26
> > > **Official Comment by Reviewer fxGj**
> > >
> > > Thank you for the authors' detailed responses and clarifications. Based on their explanations, I now have a clearer understanding of the relationships and differences between the several tables in the paper, and I acknowledge that the experimental design is relatively comprehensive.
> > >
> > > However, my fundamental concerns regarding the core innovation of this work—namely, the "U-shaped placement" of documents—remain largely unaddressed. While I agree with the authors' clarification on how their method differs from existing work, I still find the innovation depth somewhat limited. The primary contribution appears to be the application of a relatively straightforward strategy to the document task. Although the experimental results show some positive effect, the performance improvement is not substantial, which undermines the persuasiveness and impact of the proposed method.
> > > Consequently, my current assessment is that the paper still falls marginally below the acceptance threshold.
> > >
> > > To enhance the contribution of this work, I would suggest the authors consider a more promising direction: shifting the focus of the "U-shaped placement" from the entire *document* to a finer granularity of *information*. Such an exploration might more deeply uncover the potential of structured layout in information processing and could potentially lead to more groundbreaking findings.

---

### Official Review · Reviewer_7FAP · 2025-11-02

**Soundness:** 2
**Presentation:** 2
**Contribution:** 2
**Rating:** 4
**Confidence:** 3

**Summary:**

This paper studies the U-shaped positional bias of large language models (LLMs) — their tendency to attend more strongly to tokens at the beginning and end of a prompt while neglecting the middle (“lost in the middle” phenomenon). The authors analyze this behavior from both horizontal (input-level) and vertical (layer-level) perspectives using attention weight analysis.

Building on these insights, they propose a U-shaped Placement strategy that leverages positional bias by placing documents according to both their relevance and the attention distribution of the model. The first LLM pass estimates document relevance via attention weights and extracts positional bias from attention patterns. In the second pass, documents are rearranged to align important ones with high-attention positions (front and back of the prompt).

Experiments on multi-document QA benchmarks (HotpotQA, Musique, 2WikiMHQA) and multiple LLMs (Vicuna-7B, Llama-3.1-8B, Qwen2.5-7B) show consistent improvements over baselines such as RankGPT, ICR, and SELFELICIT without any additional training.

**Strengths:**

+ Turning positional bias into a benefit by explicitly optimizing for it is creative and conceptually appealing.

+ The paper systematically confirms the U-shaped bias across models and layers using both horizontal and vertical attention analysis (Fig. 2), adding interpretability to a known but underexplored phenomenon

+ The proposed algorithm requires no retraining or model modification, making it simple to deploy across different open-source LLMs and datasets.

+ The method achieves consistent gains across datasets and models (up to +5–10% EM improvement) over comparable two-pass baselines

.

**Weaknesses:**

- While the empirical motivation is solid, the paper lacks a formal justification or deeper theoretical model for why the U-shaped bias emerges and persists across architectures.

- The method assumes that attention weights correlate well with relevance, which has been debated. Without cross-validation (e.g., gradient-based attribution or causal probes), this assumption may not fully hold.

- The approach requires two LLM inference passes. Although cheaper than fine-tuning, it doubles latency and inference cost, which may matter for real-time systems.

- All datasets are QA benchmarks. It remains unclear whether the approach generalizes to non-QA tasks (e.g., summarization, reasoning, dialogue) that also suffer from “lost in the middle.”

- Gains vary across models. More analysis on model scaling or prompt length sensitivity would strengthen generality claims.

**Questions:**

1) How robust is attention as a measure of document importance? Could you provide any validation—e.g., correlation with relevance scores from gradient-based attribution, SHAP, or token perturbation tests—to support the assumption that attention weights reflect semantic importance rather than just syntactic salience?

2) If attention occasionally misaligns with relevance (e.g., focusing on stopwords or question tokens), how does that affect the placement strategy? Is there any mechanism to filter or normalize attention signals before using them for ranking?

3) Could you clarify how horizontal and vertical attention distributions are aggregated across layers and heads?
Are all attention heads equally weighted, or are only a subset of “dominant” heads used?

---

> ### Author Response · Authors · 2025-11-21
> **Response to Reviewer 7FAP: Questions**
>
> Thank you for your valuable feedback and constructive suggestions. We are committed to addressing these points to improve the quality of the paper.
>
> **R Q1: How robust is attention as a measure of document importance?**
>
> Thanks for your question. First, we would like to demonstrate that **the meaning of attention weight**. The value of attention weights reflects the influence of context tokens on generated tokens from token-level. Higher weight values indicate that the given token plays a more significant role in the computation of subsequent generated tokens, which is precisely the fundamental purpose behind the design of the entire attention mechanism.  By performing document-level aggregation, we obtain the model's internal estimation of document importance. **This value reflects the model's overall estimation of the document, influenced by numerous factors, such as the actual relevance of documents, internal biases within the model, and so on.** It may disalign with actual relevance, but we focus on separating and leveraging the influence of positional bias from attention weights in this paper. **The placement method we propose can be combined with other relevance estimation methods, such as retrieval ranking(see table 3). Document ranking based on attention weights is not the primary focus of this paper; rather, it is introduced as an optional supplementary method.**
>
> Second, **although the value of the attention weight is influenced by multiple factors, it should exhibit a monotonically increasing relationship with document relevance.**  Previous studies have explored this attention pattern and the application of its weights, such as [1], [2], and [3]. Since this is not the focus of this paper, we only demonstrate its practical performance through experimental results of different aggregation methods (see table 4).
>
> References:
>
> [1] Attention sorting combats recency bias in long context languagemodels.
>
> [2] Found in the Middle: Calibrating Positional Attention Bias Improves Long Context Utilization
>
> [3] Scalable In-context Ranking with Generative  Models
>
> **R Q2: How does the misalignment affect the placement strategy?**
>
> Thanks for your question. The attention weights we aggregate may be significantly influenced by a few high-weight tokens in the sequence. When tokens lacking semantic information become decisive, this can lead to the issue you mentioned: attention occasionally misaligns with relevance. This will affect the attention-weighted ranking but not the placement strategy. Since the position scores we obtain are aggregations of meaningless tokens, they inherently represent positional information rather than semantic information. At the same time, we conducted statistical analysis and found that the phenomenon of stopword tokens dominating the weight values did not occur in our experiments. Filter-based integration based on the actual meaning of tokens may ensure this and improve ranking accuracy, but it would further increase computational costs and is not the focus of our paper.
>
> **R Q3：How horizontal and vertical attention distributions are aggregated across layers and heads?**
>
> Thanks for your question. As shown in formula 1 and 2, we aggregate the attention weights of context tokens for the generated answer tokens. In a horizontal analysis, we compare how scores change under different inputs(ordered and unordered). The aggregation scope encompasses all layers and all attention heads, yielding an average with equal weights.  In vertical analysis, we fix the input and examine the impact of different layers. The selected heads were all averaged with equal weights.

---

> > ### Author Response · Authors · 2025-11-21
> > **Response to Reviewer 7FAP: Weakness**
> >
> > **R W1：Why the U-shaped bias emerges and persists across architectures?**
> >
> > Thanks for your question. The emergence of position bias involves factors such as the model's internal attention mechanism, hidden states and positional encoding. To thoroughly investigate its causes, it may refer to comparative experiments that evaluate different model architectures, attention designs, and positional encoding approaches. Due to computational resource constraints, we are unable to conduct such experiments. We will continue to monitor similar research. In this paper, we explore the characteristics of positional bias and propose ways to leverage it, even without knowing its true underlying mechanisms.
> >
> > **R W2: Whether attention weights correlate well with relevance**
> >
> > See *RQ1* and *RQ2*.
> >
> > **R W3: Inference Cost**
> >
> > Thanks for your question. The existing framework of our method is the two rounds of iteration of the model, which can get better results than the other two rounds of iteration methods. We will also think about how to reduce inference consumption and achieve improvement in a single pass.
> >
> > **R W4：Limited to QA Datasets**
> >
> > Thanks for your question. Our primary research focus is on enhancing the capabilities of downstream RAG question-answering systems. The method we proposed can be applied to non-QA tasks (e.g., summarization, reasoning, dialogue). We will expand other tasks when we have sufficient time in the future.
> >
> > **R W5：The Impact of Prompt and Model Size**
> >
> > Thanks for your question.  In our experiments, we employed the same prompt as in prior work to eliminate its influence and ensure a fair comparison. Although computational resource constraints prevented us from testing on larger models (70B), we conducted additional experiments on a 13B model to demonstrate the generalization capability of our approach.
> >
> > |vicuna-13b|H-unordered|M-unordered|W-unordered|H-ordered|M-ordered|W-ordered|
> > |---|---|---|---|---|---|---|
> > |vanilla|0.386|0.264|0.442|0.406|0.313|0.482|
> > |RankGPT|0.388|0.303|0.438|0.408|0.301|0.456|
> > |AttentionSort|0.41|0.305|0.458|0.408|0.301|0.474|
> > |ICR|0.406|**0.313**|0.479|0.405|0.343|0.475|
> > |SELFELICIT|0.382|0.294|0.447|0.402|0.334|0.478|
> > |Our|**0.42**|0.311|**0.484**|**0.414**|**0.357**|**0.497**|
> >
> > The results are similar to those in Table 1, demonstrating that **our method outperforms the previous baseline on the 13B model** and enhances the model's ability to utilize documentation.

---

### Note · Authors · 2025-12-03

I have read and agree with the venue's withdrawal policy on behalf of myself and my co-authors.